# Multi-Objective Optimization for Grinding of AISI D2 Steel with Al_2_O_3_ Wheel under MQL

**DOI:** 10.3390/ma11112269

**Published:** 2018-11-13

**Authors:** Aqib Mashood Khan, Muhammad Jamil, Mozammel Mia, Danil Yurievich Pimenov, Vadim Rashitovich Gasiyarov, Munish Kumar Gupta, Ning He

**Affiliations:** 1College of Mechanical and Electrical Engineering, Nanjing University of Aeronautics and Astronautics, Nanjing 210016, China; dr.aqib@nuaa.edu.cn (A.M.K.); engr.jamil@nuaa.edu.cn (M.J.); drnhe@nuaa.edu.cn (N.H.); 2Mechanical and Production Engineering, Ahsanullah University of Science and Technology, Dhaka 1208, Bangladesh; mozammelmiaipe@gmail.com; 3Department of Automated Mechanical Engineering, South Ural State University, Lenin Prosp. 76, Chelyabinsk 454080, Russia; 4Department of Mechatronics and Automation, South Ural State University, Lenin Prosp. 76, Chelyabinsk 454080, Russia; gasiyarovvr@gmail.com; 5Department of Mechanical Engineering, Chandigarh University, Gharuan 140413, Punjab, India; munishguptanit@gmail.com

**Keywords:** minimum quantity lubrication, surface grinding, multi-objective optimization, grey relational analysis, surface topography, sustainable machining

## Abstract

In the present study, the machinability indices of surface grinding of AISI D2 steel under dry, flood cooling, and minimum quantity lubrication (MQL) conditions are compared. The comparison was confined within three responses, namely, the surface quality, surface temperature, and normal force. For deeper insight, the surface topography of MQL-assisted ground surface was analyzed too. Furthermore, the statistical analysis of variance (ANOVA) was employed to extract the major influencing factors on the above-mentioned responses. Apart from this, the multi-objective optimization by Grey–Taguchi method was performed to suggest the best parameter settings for system-wide optimal performance. The central composite experimental design plan was adopted to orient the inputs wherein the inclusion of MQL flow rate as an input adds addition novelty to this study. The mathematical models were formulated using Response Surface Methodology (RSM). It was found that the developed models are statistically significant, with optimum conditions of depth of cut of 15 µm, table speed of 3 m/min, cutting speed 25 m/min, and MQL flow rate 250 mL/h. It was also found that MQL outperformed the dry as well as wet condition in surface grinding due to its effective penetration ability and improved heat dissipation property.

## 1. Introduction

AISI D2 (ENX160CrMoV12) tool steel is regarded as a key material in high performance engineering applications such as in the mold-and-die industry; for use as industrial cutting tools, gauges, and machine parts exposed to wear and injection screws; in aerospace and automotive industries; medical appliances; heavy engineering; and tools in the manufacturing industry [1]. This is due to its superior properties like high wear resistance, compressive strength, temperature resistance, and narrow tolerances as well as its high strength-to-weight ratio. However, the manufactured parts demand for high geometrical and dimensional intricacy—which raises the necessity of using a grinding operation.

Dry grinding is generally applied due to its cost-effectiveness and environmentally-friendly nature with no pollution with cost-saving lubricants. However, it involves major problems i.e., thermal damage, high friction, high residual stress, and high wheel-wear [2]. Moreover, the thermal adversities influence the integrity of the newly generated surface. For instance, as reported by Guo et al. [3], the dry surface grinding has induced a heat affected zone associated with oxidation and cutting marks. Thereby, the alternative(s) to dry cutting is being sought. Conventional flood cooling assisted cutting apparently solved this problem—but raised ecological and economic concerns. This paradoxical state urged for novel and effective solution such as minimum quantity lubrication (MQL).

The flood cutting removes the heat from the cutting zone i.e., wheel-workpiece interface, however, it was not always found to be effective. Moreover, it consumes a vast amount (i.e., 8 L per minutes [3]) of coolant/lubricant, raising a serious question about its economic and environmental sustainability. Howes et al. [4] worked on the environmental effect of grinding fluids and reported that grinding fluids have significant impact on the ecology and internal environment (workshop). Use of petroleum-based lubricants in the manufacturing sector is increasing tremendously with 1% of annual increment, and it is equivalent to 13,726 million tons of oil [5].

In MQL the fluid is delivered in extremely small quantities (10,000 times less than conventional cooling) so that for all manufacturing processes it resembles like dry machining. In this technology, an aerosol (oil–air mixture) is fed into interface of cutting tool and workpiece. This technique is free from problems like consumption of bulk quantity of fluids, their storage, and disposal. It also helps in promoting a green environment [6]. Mia et al. [7,8,9] studied the beneficial influence of MQL in the milling process, and they claimed that MQL, under different circumstances, showed better results than the dry mode of cutting. Moreover, Dhar et al. [10] have reported the superiority of MQL technology over wet cooling. Similarly, Wang et al. [11] claimed that the vegetable oil-based MQL grinding process revealed a lower specific grinding energy and friction coefficient due to its excellent lubrication property. The works of Abbas et al. [12,13] suggested application of Edgeworth–Pareto optimization of an Artificial Neural Network (ANN) to predict surface roughness (Ra) within minimal machining time (Tm), at the prime cost of machining one part (C).

The cumbersome process of hit and trial to obtain the best experimental run can be avoided by conducting systematic research optimization. Furthermore, this approach can simply identify the effect of process parameters on the ground parts for specific material. From the comprehensive literature review, summarized in Figure 1, it is discernable that researchers have studied alternative approaches of operation on various categories of material using different grinding process. The material grades are plotted on the *x*-axis, and the grinding processes on the *y*-axis. It is noted from Figure 1 that many researchers studied the effect of process parameters of the grinding process on AISI 52100, 100Cr6, Inconel, AISI 4340, AISI 8620, AISI 1045 [14], titanium alloys [15,16], and three-layer metal-composite system [17]. In the past researchers developed a model for cutting force in plunge cylindrical grinding [18], whilstin internal grinding [19], an innovate modeling approach to predict temperature distribution was developed [20]. For measurement of response Naolny et al. has proposed a new SEM-EDS-based analysis of grinding of titanium grade 5 [21]. Sutowski et al. worked on monitoring the cylindrical grinding process using acoustic emission signal (AE) and image analyses [22]. Similarly, the researchers have used the Taguchi method, response surface methodology (RSM), regression analysis as well as the conventional experiments to model responses of grinding process. Prediction of response using optimal combinations of significant process parameter has made RSM the most effective approach.

It can be concluded from the literature review that the process parameters such as workpiece speed, depth of cut, cutting speed and MQL flow rate have a direct influence on the workpiece in grinding process and effect of these process parameter varies from material-to-material. Literature review shows that no work has been reported yet on AISI D2 steel to evaluate, compare, and optimize (surface roughness, temperature, and force) using MQL grinding process. The authors get motivation by the fact that insufficient work has been done on the multi-objective optimization of grinding of AISI D2 steel. Moreover, the lubrication performance of Accu-Lube 6000 in MQL grinding has not been addressed. Therefore, this experimental investigation shows a systematic approach to evaluate the influence of above-mentioned input parameters on grinding outputs. This research aims to develop an empirical relationship using RSM to predict the normal forces, surface roughness, and temperature and to optimize the process parameters to get the best surface finish, minimum forces, and temperature.

## 2. Methodology

### Experimental Setup

The surface grinding was performed on AISI D2 steel having 1.56% carbon and 12% chromium by weight. The chemical composition of the workpiece material is verified through optical emission spectrometer before experiments. Alumina oxide grinding wheel manufactured by NORTON (Shanghai, China) was used. The grinding process was performed on a surface grinding machine with Model (WAZAA415X-NC) with motor driving having a capacity of 6 kW. Surface grinding was employed on AISI D2 steel having chemical composition as presented in Table 1. Alumina oxide grinding wheel (FE 38A60KV) manufactured by NORTON was used in present work according to the cutting condition, hardness and tool manufactures recommendation. Alumina Oxide vitrified grinding wheel of size (250 mm × 72.5 mm × 31.8 mm) was used as abrasive cutting material.

The experiments were performed under three conditions namely dry, wet, and MQL grinding. Under the dry condition, grinding was performed without coolant. Castrol Syntilo 9954 (Shangai, China) with density 1066 kg/m^3^ was used as the coolant for conventional wet grinding. A flow rate of 6 L/min was maintained throughout wet grinding. For experiments involving grinding under MQL condition Accu-Lube 6000 with specific gravity 0.92, flash point 214 °C, and density 8.9 CSt special cutting oil was used at variable flow rates between 50 and 50 mL/h. During the whole process the air pressure was 6 bar, with a nozzle angle of 15°, and nozzle distance of 30 mm, which were kept constant. The schematic diagram for surface grinding process along with responses measurement has been shown in Figure 2.

Response surface methodology (RSM) with central composite design (CCD) has been considered for the modeling and analysis of the performance measures of the ground workpiece. In CCD, if some process parameters are represented by k and number of central points by m, the total number of experimentation can be calculated by Equation (1) [53]. Process parameters and their levels for Dry, Wet, and MQL grinding are shown in Table 2.
(1)$$\mathrm{Number}\;\mathrm{of}\;\mathrm{Experiments}\;=\;2k\;+\;2^{k}\;+\;m$$

A total of 30 experiments have been performed for different process parameter combinations (depth of cut, table speed, cutting speed, and lubricant flow rate) in MQL-grinding and 18 experiments have been performed with process parameter (depth of cut, table speed, and cutting speed) for the dry and wet grinding processes. The experimental run, process parameters, and response parameters have been shown in Table 3 and Table 4.

## 3. Results and Discussion

In this section, the normal force, grinding temperature, and surface roughness are analyzed by the experimental investigation, statistical analysis, mathematical modeling, and graphical plots.

### 3.1. Normal Forces

The cutting forces largely influence the performance of grinding. The grinding mechanics are, in turn, altered by the lubrication effects prevailing between the wheel and work surface. The normal forces were measured during the grinding process using dynamometer. To check the influence of dry, wet, and MQL conditions on the normal forces, a comparison has been drawn in Figure 3. Owing to the dominant role of depth of cut on normal forces (discussed later), the comparison has been performed against all combinations of depth of cut, at different values of cutting speed, and table speed. This comparison shows variation in the normal force with the variation in the cutting parameters. Note that a comparison has been performed for a set of seven experiments where cutting conditions were same in each environment, and cutting conditions have been selected from design space provided in Table 3 and Table 4. The normal force in dry grinding was kept as a reference point to calculate percent reduction.

It can be seen that a reduction of 37.86 to 79.26% in normal forces was found for MQL grinding; however, a reduction of only 22.33 to 57.27% in forces was achieved for wet grinding. It can be concluded from the results that the MQL environment has generated lower forces than the dry and wet environment.

MQL-assisted grinding confirms effective supply of the mist of oils (droplets) in between the interfaces of workpiece-wheel grain that provides better and efficient lubrication. This efficient lubrication improves the slipping of grain between the workpiece-wheel. Penetration of small oil–mist droplets into the cutting region with high speed explains the effectiveness of the process [54]. It makes a consistent lubricant tribofilm with reduced shearing strength than the base material, and it ensures the lower normal forces in MQL-assisted grinding in comparison with the wet and dry mode.

In case of MQL assisted grinding process, the grinding forces are decreased significantly; this is due to the proper penetration of cutting fluid as in aerosol form. Durable oily films formed during MQL grinding allow proper slipping of grinding wheel grains and offer advanced level lubrication at high pressure. The second reason for MQL having least forces could be excellent and superior lubrication properties of Acculube-6000 than Syntile-9956. Moreover, the abrasive grain sharpness is maintained for a long time due to this excellent lubrication. Larger forces in dry grinding are due to the absence of cutting fluids, which leads to dulling of abrasive grits, premature breakage of grains and side flow of materials [41]. The results are in good agreement with the published literature, such as with the study of Sadeghi et al. [40]. They have also claimed to have better surface quality and lower force while grinding. This was accredited to the lower coefficient of friction.

Besides the comparison of the normal force, to make the study complete, here the contribution of the factors on normal force is evaluated by analysis of variance (ANOVA). This statistical analysis was conducted using Design Expert 8.0.7. The fit summary for normal force (dry, wet, and MQL) suggested the quadratic relation as the best fit model. In dry grinding, the main effects of depth of cut, table speed, and cutting speed were identified as the significant model terms for the force. The main parameters that contribute significantly to forces generation in wet grinding includes depth of cut and cutting speed. The ANOVA result suggested that the main effects of depth of cut, table speed, cutting speed, and MQL flow rate were the significant model terms associated with the force for MQL grinding. Model for the prediction of force in dry, wet and MQL environment are provided in Equations (2)–(4) respectively.

The ANOVA includes the significant terms along with adequacy measures *R*^2^, Adj. *R*^2^ and predicted *R*^2^ is close to ‘1’ indicating the adequacy of resulting model for all considered environments. The results of ANOVA indicate that the model is significant with a *p*-value of less than 0.05.
(2)$${\mathrm{DRY}\;}_{\left(F\right)}\;=\;+34.5\;+\;(6.4\;\times \;a_{p})\;-\;\left(7.8\;\times \;v_{w}\right)\;-\;\left(1.7\;\times \;V_{s}\right)\;-\;\left(0.16\;\times \;a_{p}\;\times \;v_{w}\right)\;-\;\left(0.33\times \;a_{p}\;\times \;V_{s}\right)\;+\;\left(0.41\;\times \;a_{p}\;\times \;v_{w}\right)\;+\;\left(0.096\;\times \;{a_{p}}^{2}\right)\;+\;\left(0.28\;\times \;v^{2}\right)\;+\;\left(0.13\;\times \;{v_{s}}^{2}\right)$$
(3)$${\mathrm{WET}\;}_{\left(F\right)}\;=\;+219.3\;-\;(7.6\;\times \;a_{p})\;-\;\left(13.09\;\times \;v_{w}\right)\;-\;(3.8\;\times v_{s})\;+\;\left(0.31\;\times \;a_{p}\;\times \;v_{w}\right)\;+\;\left(0.10\times \;a_{p}\;\times \;v_{s}\right)\;+\;\left(0.06\;\times \;v_{w}\;\times \;v_{s}\right)\;+\;(0.1050\;\times \;{a_{p}}^{2})\;+\;(0.34375\;\times \;{v_{w}}^{2})\;+\;(0.005\;\times \;{v_{s}}^{2})$$
(4)$${\mathrm{MQL}\;}_{\left(F\right)}\;=\;+95.3\;+\;\left(2.3\;\times \;a_{p}\right)\;-\;\left(1.4\;\times \;v_{w}\right)\;-\;\left(7.2\times \;v_{s}\right)\;-\;\left(0.1\;\times \;Q\right)\;-\;\left(0.01\;\times \;a_{p}\;\times \;v_{w}\right)\;+\;\left(0.03\;\times \;a_{p}\;\times \;v_{s}\right)\;-\;\left(3.6\;\times \;10^{-4}\times \;a_{p}\;\times \;Q\right)\;+\;\left(0.1\;\times \;v_{w}\;\times \;v_{s}\right)\;-\;(0.018\;\times \;v_{w}\;\times \;Q)\;+\;\left(0.004\;\times \;v_{s}\;\times \;Q\right)\;-\;\left(0.01\;\times \;{a_{p}}^{2}\right)\;+\;(0.25\;\times \;{v_{w}}^{2})\;+\;\left(0.08\;\times \;{v_{s}}^{2}\right)\;+\;\left(5.4\;\times \;10^{-4}\;\times \;Q^{2}\right)$$

Figure 4a–c demonstrates the contribution of the depth of cut and table speed on the normal force found in dry, wet, and MQL grinding. By comparing the response surface plots for dry, wet and MQL, it is evident that the behavior of force is nearly similar at varying depth of cut and table speed conditions. The force is more sensitive to variation in depth of cut compared to table speed. The similar results are also claimed in a previous paper [55]. The force increases with the increase in table speed in dry grinding; however, in wet MQL grinding table speed has a negligible effect.

### 3.2. Temperature

Temperature is produced due to the interaction of the grinding wheel with the workpiece. Data for temperatures were recorded using an Infrared thermometer (Raytek-Raynger MX4 (Guangdong, China)). Comparison of temperatures allowed us an assessment of the thermal performance of the MQL-assisted grinding compared to dry and wet cases. It gives an insightful analysis of lubrication in wet and MQL techniques. As the depth of cut was found the most significant parameter for temperature (discussed later), therefore, it is plotted on the *x*-axis in the comparison graph in Figure 5. It is depicted in Figure 4a that the temperature reduction in wet grinding is between 65.83% and 72.44% and in MQL grinding is between 55.18% and 67.4%. It is further noticeable that the temperature achieved in MQL environment is much lower than dry and close to the wet temperature; it is even lower than the flash point of oil used which recommends its usage during grinding.

Comparison of measured values of temperatures in dry, wet and MQL mode at workpiece surface is shown in Figure 4a. As expected, the wet mode results in less cutting temperature and dry mode produces the highest temperature. By applying the MQL method, the temperatures values are about 74–134 °C and this is less than those in dry grinding conditions. This is due to effective cooling and lubrication of the oil mist provided by MQL that reduces the cutting temperature. Moreover, a significant amount of heat is removed through convection heat transfer phenomena [42].

The reason for higher temperatures in dry grinding is intrinsically associated with high specific energy requirement. This high specific energy demand is due to grain sliding phenomena and shearing with adverse grit geometry. Such high temperature is usually for thermal damages on the ground surface, such as residual stress, oxidation, and burning.

From Figure 5, moderately low-temperature values have been seen in all cooling modes. However, the combined effects of lubrication (the lowest normal forces) and cooling are main reasons for the improved cutting efficiency in MQL technology. It is worth noticing that the maximum temperature achieved in MQL is far below the thermal damage temperature. The lowest temperature in the wet cooling mode could be due to the excellent cooling provided by Castrol Syntilo 9954. In addition, the obtained results are in good agreement with the previous literature survey [21].

The wet cooling eliminates the symptoms of the problems but not the cause. The MQL induces a different approach to deal with heat and friction in the machining process. In the MQL-assisted machining process, the interface of the cutting tool and workpiece is coated with the high-quality oil droplets delivered in an atomized spray. This thin layer of lubricant is extremely efficient at reducing friction and therefore cooling the temperature.

The secret of Accu-Lube 6000 success is the polar nature of its molecule. Molecules of the selected lubricant align themselves by their poles making bonds exceptionally strong, both to each other and tometallic surfaces. This strong bond allows Accu-Lube 6000 to create a tough and desirable lubrication barrier. The MQL is not only useful for lubricating the grinding wheel, but also it removes heat from the grinding zone.

For further analysis, the statistical tool ANOVA was used to check the adequacy of the temperature model in each environment. The quadratic relationship was found as the best fit model for temperature. The main and interaction effect that contributes significantly to temperature for dry grinding includes depth of cut, table speed, and cutting speed. The ANOVA for wet grinding revealed that the main effects of depth of cut and table speed were noted as significant model terms associated with temperature. On the other hand, ANOVA result divulged that main process parameter that contributes significantly in temperature for MQL grinding includes depth of cut, table speed, cutting speed, and MQL flow rate. The ANOVA results of temperature for dry, wet, and MQL can be validated from Table 3 and Table 4. Finally, the mathematical models for temperature in dry, wet and MQL environment are provided in Equations (5)–(7) respectively.
(5)$${\mathrm{DRY}\;}_{\left(T\right)}\;=\;+445.08\;-\;\left(7.4\;\times \;a_{p}\right)\;-\;(26.2\;\times \;v_{w})\;-\;(6.4\;\times \;v_{s})\;+\;(0.4\;\times \;a_{p}\;\times \;v_{w})\;+\;(0.19\;\times \;a_{p}\;\times \;v_{s})\;+\;(0.08\;\times \;{a_{p}}^{2})\;+\;(1.28\;\times \;{v_{w}}^{2})\;+\;(0.05\;\times \;{v_{s}}^{2})$$
(6)$${\mathrm{WET}\;}_{\left(T\right)}\;=\;+235.9\;-\;\left(8.82\;\times \;a_{p}\right)\;+\;(1.46\;\times \;v_{w})\;-\;(7.5\;\times \;v_{s})\;-\;(0.31\;\times \;a_{p}\;\times \;v_{w})\;+\;(0.19\;\times \;a_{p}\;\times \;v_{s})\;-\;(0.01\;\times v_{w}\;\times \;v_{s})\;+\;(0.16\;\times {a_{p}}^{2})\;+\;(0.56\;\times \;{v_{w}}^{2})\;+\;(0.060\;\times \;{v_{s}}^{2})$$
(7)$${\mathrm{MQL}\;}_{\left(T\right)}\;=\;+53.8\;+\;\left(2.4\;\times \;a_{p}\right)\;+\;(18.9\;\times \;v_{w})\;-\;(3.5\;\times \;v_{s})\;-\;(0.03\;\times \;Q)\;-\;\left(0.02\;\times \;a_{p}\;\times \;v_{w}\right)\;-\;(0.02\;\times \;a_{p}\;\times \;v_{s})\;-\;(0.004\;\times \;a_{p}\;\times \;Q)\;+\;\left(0.03\;\times \;v_{w}\;\times \;v_{s}\right)\;-\;(0.015\;\times \;v_{w}\;\times \;Q)\;+\;\left(0.002\;\times \;v_{s}\;\times \;Q\right)\;+\;\left(0.003\;\times \;{a_{p}}^{2}\right)\;-\;(1.1\;\times \;{v_{w}}^{2})\;+\;\left(0.07\;\times \;{v_{s}}^{2}\right)\;-\;(3.16\;\times \;10^{-4}\;\times \;Q^{2})$$

Furthermore, 3D response surface graphs in Figure 6a–c highlights the effects of depth of cut and table speed on temperature for dry, wet, and MQL environments. It is noted that the interaction effect of depth of cut and table speed is not similar for all given environment. In the case of dry environment (Figure 6a), the temperature increases as the depth of cut increases; also the depth of cut has a direct and significant effect on temperature. However, the table speed has a less significant effect on temperature compared to the effect of depth of cut; therefore, the temperature slightly increases with an increase in table speed. Figure 6b depicts that in wet grinding the temperature increases rapidly at a higher level of table speed. Figure 4c represents the 3D response surface plot for MQL grinding; in that case, the MQL flow rate was the most significant parameter for temperature control. It can be seen that initially the temperature decreases with an increase in MQL flow rate (Q). The minimum temperature was found at the lowest level of MQL flow (Q).

### 3.3. Surface Roughness

Surface roughness is an important parameter to analyze the quality of the surface of the workpiece. The surface finish largely influences the dimensional accuracy and product life. Surface roughness values were measured by using portable stylus profilometer Surftest SJ-410 (Mitutoyo, Tokyo, Japan). To elaborate the effectiveness of MQL over the wet and dry environment, a comparison of surface roughness in three different environments has been made and presented in Figure 7. Since the table speed was found as the most significant parameter for surface roughness (discussed later), therefore, the comparison has been drawn by varying the table speed and keeping the other process parameters at a constant level. During comparison, the dry grinding process was set as a reference point. It can be noted from Figure 7 that wet grinding resulted in a 42 to 69.7% reduction in surface roughness, whereas MQL grinding was only able to reduce roughness by 32 to 66.1% when compared with the dry grinding process. From these results, it is clear that MQL provides better results than dry and almost similar results with wet conditions.

The better surface roughness generated by MQL is because of excellent control of temperature in grinding zone. An increase in the depth of cut increases grinding temperature which softens the workpiece material. This causes the clogging which produces low-quality surfaces with high surface roughness. In dry grinding, due to the absence of cutting fluid the frictional forces increase and these frictional forces are converted to heat which is further transferred to workpiece material and chips. This phenomenon influences the surface roughness, and it is responsible for surface burns and grinding mark on the workpiece. In wet grinding, these defects are avoided by the intensive lubrication.

In the dry mode, there is no cooling, hence, even at a depth of cut 25 µm, burning traces started to appear. This phenomenon is called grain flattening, and it results in high surface roughness and thermal damages of the ground surface as compared to other cooling modes. It is noted from Figure 7 that under specific conditions MQL competes with, and even performs better than, the wet cooling mode. Moreover, the results achieved are in good agreement with the published literature [21].

From Figure 7, it can be seen that the wet cooling mode provides better surface finish than MQL. The reason for this lies in the fact that when MQL was applied in machining of hardened steel, the frictional coefficient decreased; however, an increase in material deformation leads to poor surface quality. Furthermore, during MQL mode, the lower normal forces have been observed which reduced the self-sharpening of grinding wheel. Hence, a total number of cutting edges and their sharpness are lower than the wet cooling mode that leads to thicker chips and relatively high surface roughness values. These experimental results are in good agreement with the published literature [29].

For further analysis, ANOVA was performed, and it revealed that the main effects of table speed, and cutting speed were found as significant model terms associated with surface roughness for dry grinding. The main influence includes the effects of table speed, cutting speed, depth of cut, cutting speed, and table speed as the significant model terms associated with surface roughness. The empirical models for prediction of surface roughness under dry, wet and MQL grinding are provided in Equations (8)–(10), respectively.
(8)$${\mathrm{Dry}}_{\left(\mathrm{Ra}\right)}\;=\;–8.13\;+\;\left(0.25\;\times \;a_{p}\right)\;+\;(0.66\;\times v_{w})\;+\;(0.24\;\times \;v_{s})\;+\;(0.0037\;\times \;a_{p}\;\times \;v_{w})\;-\;(0.0019\;\times \;a_{p}\;\times \;v_{s})\;-\;(0.011\;\times \;v_{w}\;\times \;v_{s})\;-\;(0.004\;\times {a_{p}}^{2})\;-\;(0.023\;\times \;{v_{w}}^{2})\;-\;(0.001\;\times \;{v_{s}}^{2})$$
(9)$${\mathrm{Wet}}_{\left(\mathrm{Ra}\right)}\;=\;-0.55\;-\;\left(0.02\;\times \;a_{p}\right)\;+\;(0.16\;\times v_{w})\;+\;(0.038\;\times \;v_{s})\;+\;(7.5\;\times \;10^{-4}\times \;a_{p}\;\times \;v_{w})\;-\;(0.001\;\times \;a_{p}\;\times \;v_{s})\;-\;(0.005\;\times \;v_{w}\;\times \;v_{s})\;+\;(0.001\;\times \;{a_{p}}^{2})\;+\;(0.002\;\times \;v^{2})\;+\;(5.9\;\times \;10^{-4}\;\times \;{v_{s}}^{2})$$
(10)$${\mathrm{MQL}}_{\left(\mathrm{Ra}\right)}\;=\;+0.52\;-\;\left(0.014\;\times \;a_{p}\right)\;+\;(0.028\;\times \;v_{w})\;-\;(0.014\;\times \;v_{s})\;-\;\left(6.78\;\times \;10^{-4}\times \;Q\right)\;-\;\left(4.6\;\times \;10^{-4}\;\times \;a_{p}\;\times \;v_{w}\right)\;+\;(5.1\;\times \;10^{-4}\;\times \;a_{p}\;\times \;v_{s})+\;(2.8\times \;10^{-5}\;\times \;a_{p}\;\times \;Q)\;-\;(5.38\;\times 10^{-4}\;\times \;V_{w}\;\times \;V_{s})\;+\;(2.8\;\times \;10^{-5}\;\times \;v_{w}\;\times \;Q)\;-\;\left(3.8\;\times \;10^{-5}\;\times \;v_{s}\;\times \;Q\right)\;+\;\left(1.3\times \;10^{-4}\;\times \;{a_{p}}^{2}\right)\;+\;(0.003\;\times \;{v_{w}}^{2})\;+\;(2.8\;\times \;10^{-4}\;\times \;{v_{s}}^{2})\;+\;\left(1.3\;\times \;10^{-6}\;\times \;Q^{2}\right)$$

It is significant to know that Figure 8a–c shows the effect of two process parameters (simultaneously) on the output parameters while other process parameters are fixed at their middle levels. The effects of table speed and cutting speed on surface roughness (*R_a_*) for dry, wet, and MQL grinding environments are presented in Figure 8a–c, it is evident from these figures that the surface roughness has a direct relation to table speed and cutting speed, i.e., an increase in speed also increases surface roughness. The impact of changing table speed is more at a low level of cutting speed as compared to the high level of cutting speed. Similarly, the impact of table speed on surface roughness is higher at a low level of table speed as compared to the high level of table speed.

### 3.4. Surface Topography for MQL-Assisted Grinding

Appropriate analysis of surface components, in terms of waviness, form, and roughness and multi-scalar feature topographical features, is very important. Surface topography was studied using noncontact 3D surface profilometer (S NEOX) with magnification ×1000 for AISI D2 steel in the MQL-assisted grinding process (Figure 9a). Figure 9b,c highlights 2D and 3D topography. The surface topography area was considered for 2mm along the grinding width direction and 1.5 mm along the grinding direction. From the experimental data, it was found that the table speed has a significant influence on ground surface roughness and topography. Figure 9 shows the machined surface topography of the ground surface at depth of cut of 15 µm, table speed of 3 m/min, cutting speed of 25 m/s, and MQL flow rate of 250 mL/h; this was obtained using SENSOFAR with magnification ×1000 for AISI D2 steel in the MQL-assisted grinding process. Figure 9 indicates that in the MQL-assisted grinding process the plastic deformation by the mechanical load, distortion of the surface layer, and pull-out of grains are not present on the ground surfaces. MQL technique has fewer defects, especially no plastic deformation, due to higher lubrication conditions. It was noted that in MQL-assisted grinding a minimum of 0.31 µm surface roughness was achieved.

## 4. Formulation and Validation of Multi-Objective Optimization

Due to superior nature of MQL-grinding, as delineated earlier, compared to dry as well as conventional wet grinding condition, the optimization is performed only for MQL-grinding. Amulti-objection optimization was performed to concurrently optimize the normal force, surface roughness, and temperature to select optimum cutting parameters in MQL-assisted grinding. It is pertinent to mention that, to avoid over presentation of the data and calculation, only the ANOVA for the Grey Relational Grade (GRG) has been displayed.

### 4.1. Evaluation of Optimal Parameters for MQL Grinding

Single response optimization is simple to perform and commonly used to address the problems for optimization. However, that method cannot be used to solve multiple-criteria decision-making problems [6]. Therefore, to solve this problem, in this study the response surface methodology (RSM)-based grey relational analysis (GRA) method has been deployed for multiresponse optimization. Grey relational analysis consists of three simple steps: (a) Normalizing measured data for surface roughness, temperature, and force (minimum the better) using Equation (11); (b) calculation of Grey relational coefficient (GRC) using Equation (12); and (c) finally, computing grey relational grade based on equal weights using Equation (13). The detailed method of calculating grey relational coefficient (GRC) and grey relational grade (GRG) can be seen from previous works [41,55].
(11)$$x_{i}^{*}\left(k\right)=\;\frac{\max\left(x_{i}^{0}\right)-x_{i}^{0}\left(k\right)}{\max(x_{i}^{0}\left(k\right)-\min\left(x_{i}^{0}\left(k\right)\right)}$$
(12)$$\xi \left(k\right)=\;\frac{{∆}_{min}+\xi {∆}_{max}}{∆0_{i}\left(k\right)+\xi {∆}_{max}}$$
(13)$$\gamma_{i}=\frac{\sum_{k=1}^{n}w_{k}\xi_{i}\left(k\right)}{n}$$

Here, ∆0*_i_*(*k*) is known as deviation sequence and it is the absolute value. The minimum and maximum values of ∆0*_i_*(*k*) are denoted by ${∆}_{min}$ and ${∆}_{max}$ respectively. $\gamma_{i}$ represents the grey relational grade of each experiment. The values of ξ lies between 0 and 1 and it is called distinguishing coefficient, normally its value is taken as 0.5. $w_{k}$ represents the normalized weight of factor k. Table 5 consists of GRC and GRG of MQL grinding.

### 4.2. ANOVA for Grey Relational Grade

ANOVA results give useful information about significant grinding parameters at the 5% significance level (95% confidence level). As listed in Table 6, ANOVA has been carried out to analyze the effect of grinding parameters on multiresponses.

The results show the highest influence was exerted by the table speed on GRG. Moreover, the *p*-value (0.0001), which is less than 0.05, shows the significance of the model. For GRG, the main effects of depth of cut, table speed, and MQL flow rate; interaction effects of depth of cut and cutting speed ($a_{p}$ × $v_{s}$), table speed, and cutting speed ($v_{w}$ × $v_{s}$); depth of cut, MQL flow rate $\left(a_{p}\times Q\right)$, table speed, and MQL flow rate $v_{w}\times Q$; and quadric effects of cutting speed ($v_{s}$^2^) and table speed (${v_{w}}^{2})$, were identified as the significant model terms. The ANOVA results comprising of significant model terms along with adequacy measures *R*^2^, adjusted *R*^2^ and predicted *R*^2^ are shown in Table 6. The values of adequacy measures *R*^2^, adjusted *R*^2^ and predicted *R*^2^ close to 1 indicate the adequacy of the resulting models. The empirical models for the prediction GRG has been provided in Equation (14).
(14)$${\mathrm{MQL}}_{\left(\mathrm{GRG}\right)}\;=\;+0.51\;-\;(0.045\;\times \;a_{p})\;-\;(0.11\;\times v_{w})\;-\;(6.8\times \;10^{-3}\;\times \;v_{s})\;+(0.062\;\times \;Q)\;+\;\left(0.018\;\times \;a_{p}\;\times \;v_{w}\right)\;-\;(0.015\;\times \;a_{p}\;\times \;Q)\;-\;\left(0.014\;\times \;V_{w}\;\times \;Q\right)\;-\;(4.2\times \;10^{-3}\;\times \;{a_{p}}^{2})\;+\;(0.036\;\times \;{v_{w}}^{2})\;-\;(0.014\;\times \;{v_{s}}^{2})\;+\;\left(6.4\;\times \;10^{-3}\;\times \;Q^{2}\right)$$

### 4.3. Confirmatory Test

The statistical model has been employed to verify the adequacy of the developed model. In the final step of RSM, to validate the model experimentally, confirmation experiments have been performed using RSM based grey relational analysis GRA technique. For the determination of improvement in responses and to validate the accuracy of optimization, confirmation experiments were employed at initial levels (experiment 19) and optimal levels. These experiments were repeated three times to mitigate the error. It is relevant to mention that multi-objective optimization and confirmation experiments are performed for all environments, however, Table 5 consist of data only for MQL environments. The estimated grey relational grade, which represents optimal cutting conditions, can be computed as follows,
(15)$$\gamma_{p}=\gamma_{m}+\sum_{i=1}^{k}\left(\gamma_{i}-\gamma_{m}\right)$$
where, $\gamma_{p}$ is a predicted grey relational grade, $\gamma_{m}$ is a total mean of GRG, and $\gamma_{i}$ is an optimum average GRG value at specific cutting parameter level, *i* = 1, 2, …, *k*, where ‘*k*’ is the number of parameters significantly affect the GRG [56].

It was noted from Table 7 that percentage improvement in the GRG from initial cutting conditions ($a_{p}$^3^-$v_{w}$^1^-$v_{s}$^3^-Q^3^) to the optimum cutting condition ($a_{p}$^1^-$v_{w}$^1^-$v_{s}$^3^-Q^5^) was 3.18 and these results are in good agreement with the results from the literature. Pawade and Joshi [55] employed the Taguchi grey relational analysis (TGRA) method in the turning process and they achieved 4.11% achievement in optimum GRG when compared with initial conditions.

## 5. Conclusions

In this paper, the effects of the cutting parameter and the cooling mode on the grinding zone temperature, normal forces, and ground surface quality of AISI D2 steel were investigated. The influence of process parameter on three different responses has been analyzed based on the developed mathematical model. A mathematical model for each response in each cooling mode has been developed to correlate the process parameter. Finally, multi-objective optimization was performed for MQL cooling mode. Based on the results of the experimental investigation, the following comparative conclusions for MQL grinding can be drawn.
In all grinding conditions, the surface roughness is profoundly affected by the table speed, and the surface roughness values achieved in wet and MQL conditions are comparable.Depth of cut has been found as the critical process parameter for temperature in all cutting conditions. Comparison results for temperatures show that overall minimum temperature was achieved in wet grinding followed by MQL-assisted grinding. Moreover, it was noted that MQL grinding results in a temperature reduction of 55.18 to 67.4% as compared to dry grinding.Cutting forces were found increasing as the depth of cut was increased. Comparative results for forces have shown that minimum cutting forces were achieved in MQL environment. When compared with dry grinding, 37.86–79.26% reduction in forces was found in MQL-assisted grinding.During the experiments, it was observed that MQL flow rates values used in the grinding process did not cause in the dispersion of mist, which is environmentally friendly. More interesting to mention, the MQL maintained the cutting zone clean like dry grinding, and it produced a comparable, and sometimes better, result when compared to wet grinding.The modeling and optimization of the MQL-assisted grinding process can be applied in the green manufacturing process. It was noted that RSM prediction is well matched with the experimental results. Also, research findings and a mathematical model developed in this study can be used in the industry by the machinist to predict the surface roughness, temperature, and normal forces for grinding of AISI D2 steel.The work surface quality in MQL is comparable to wet condition. Therefore, MQL is recommended over flood cooling due it MQL’s economic and clean production perspective. However, considerable further research is needed to fully exploit its potential.

Few works have been published to check the effectiveness of vegetable oils in MQL-assisted grinding. Use of a mixture of two or more oils or may be explored with the addition of nanoparticles, which can enhance the lubricating and oxidation stability of these oils. Still, efforts are needed to improve the surface integrity of difficult to cut materials during machining under the MQL-assisted lubri-cooling mode. Moreover, analysis of the surface integrity of tritium and Inconel alloys using hybrid lubri-cooling techniques such as MQL and LN_2_ is still not exploded very well. Finally, further research should be carried out on how these newly developed technologies can become sustainable regarding economic and environmental aspects.

## Figures and Tables

**Figure 1 materials-11-02269-f001:**
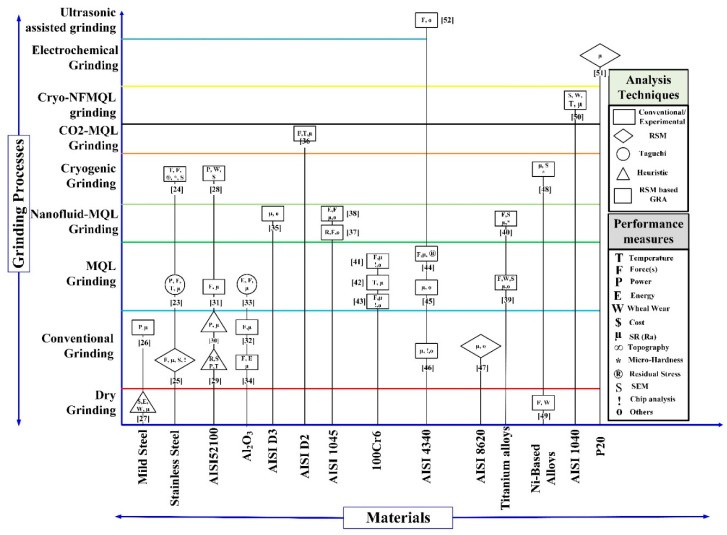
Overview of research on grinding types, the different materials used, and methods used [22,23,24,25,26,27,28,29,30,31,32,33,34,35,36,37,38,39,40,41,42,43,44,45,46,47,48,49,50,51,52].

**Figure 2 materials-11-02269-f002:**
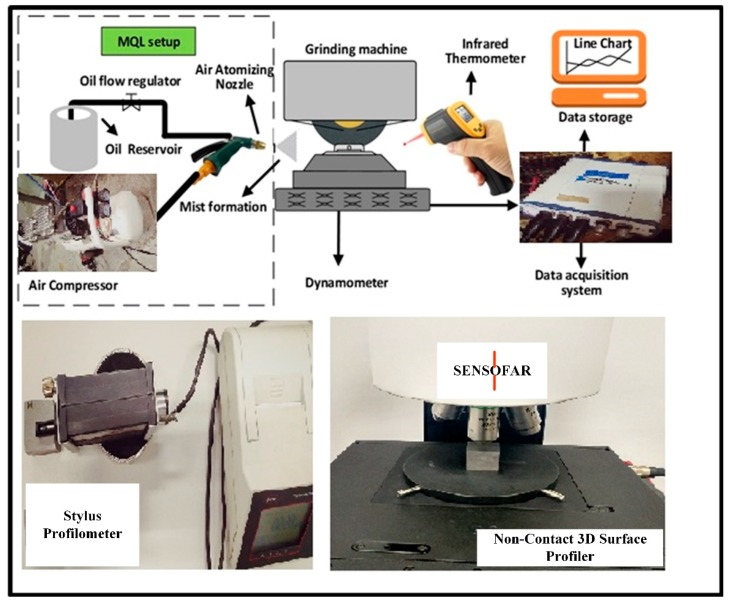
Schematic diagram showing minimum quality lubrication (MQL) working principle and response measurements.

**Figure 3 materials-11-02269-f003:**
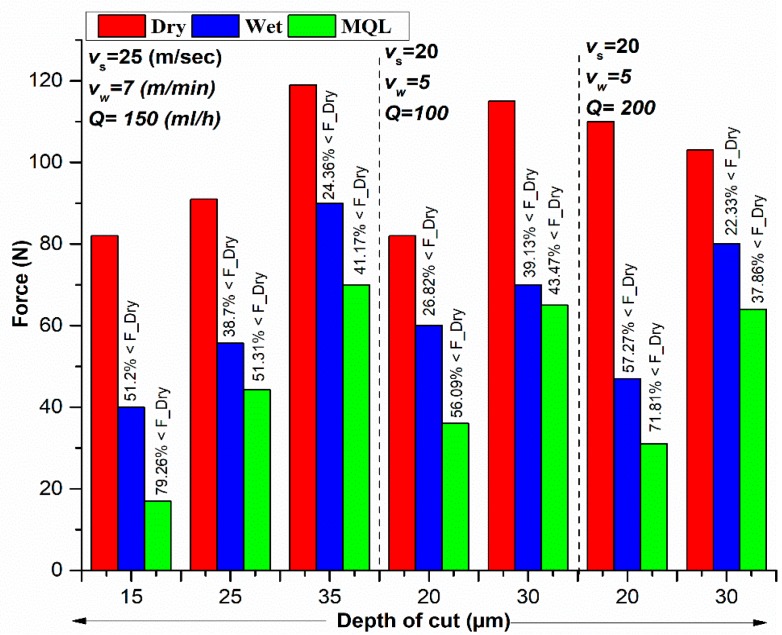
Comparison of force measurements in different cooling environments.

**Figure 4 materials-11-02269-f004:**
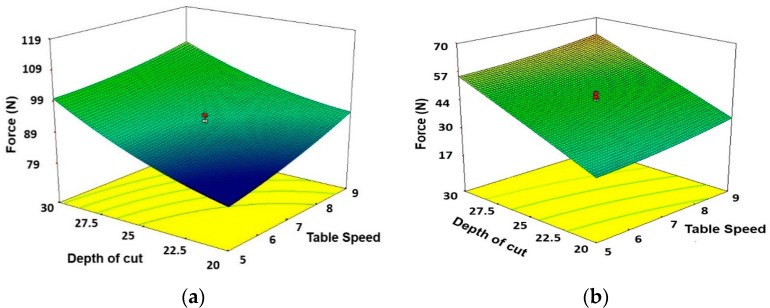
3D Surface plots showing the effects of depth of cut and table speed on forces for (**a**) Dry, (**b**) Wet, and (**c**) MQL conditions. Units of table speed and depth of cut are ‘m/min’ and ‘µm’, respectively.

**Figure 5 materials-11-02269-f005:**
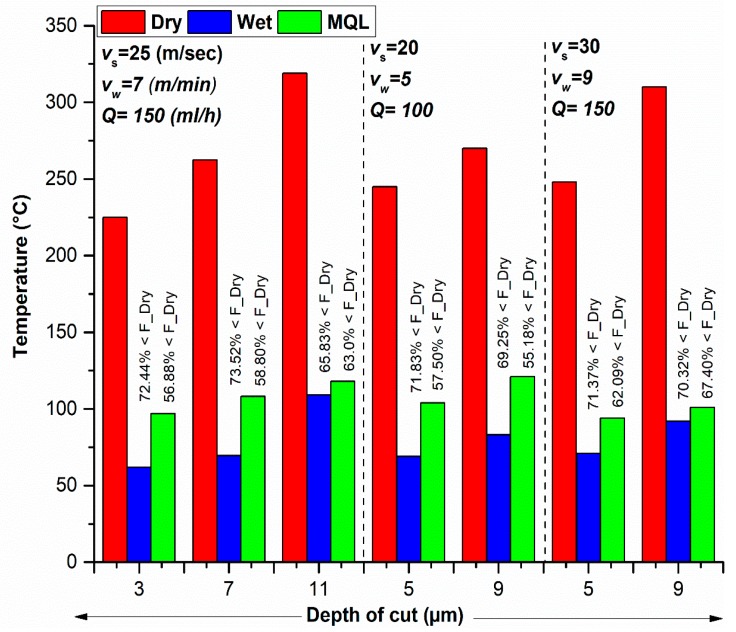
Comparison of temperature measurements in different cooling environments.

**Figure 6 materials-11-02269-f006:**
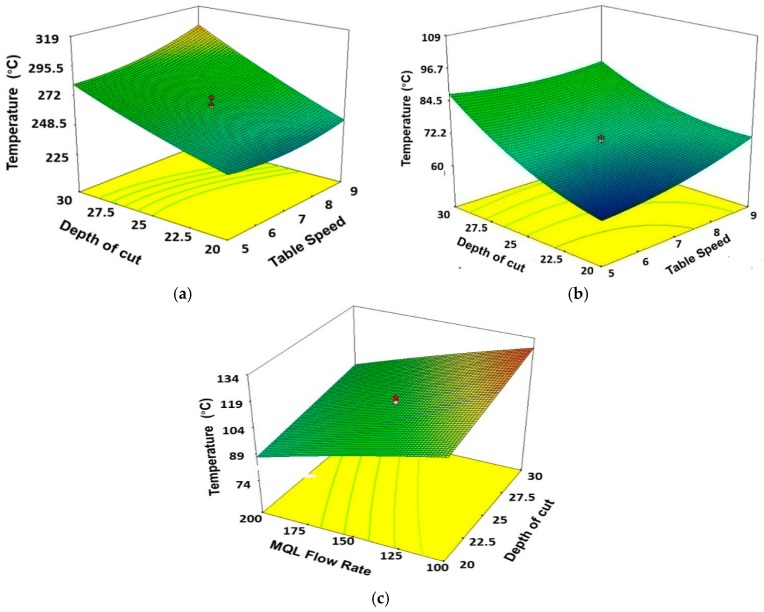
Surface plots showing the effects of depth of cut and table speed on temperature for (**a**) Dry, (**b**) Wet, (**c**) and MQL conditions. Units of table speed, depth of cut, and MQL flow rate are ‘m/min’, ‘µm’, and ‘mL/h’, respectively.

**Figure 7 materials-11-02269-f007:**
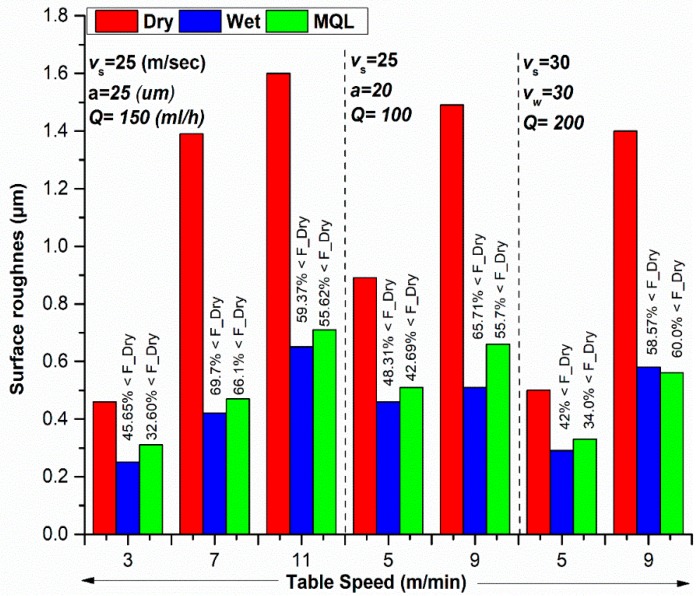
Comparison of surface roughness in different cooling environments.

**Figure 8 materials-11-02269-f008:**
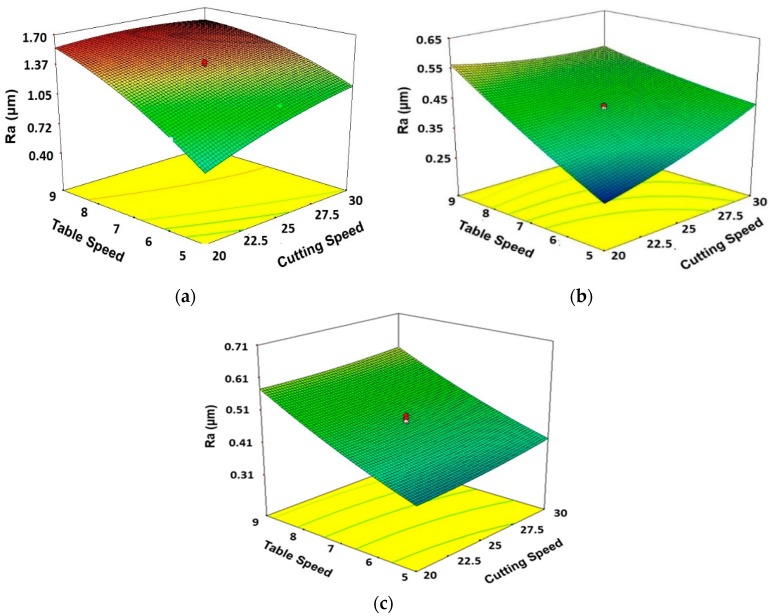
3D Surface plots showing the effects of cutting speed and table speed on surface roughness for (**a**) Dry, (**b**) Wet, and (**c**) MQL conditions. Units of table speed and cutting speed are ‘m/min’ and ‘m/s’, respectively.

**Figure 9 materials-11-02269-f009:**
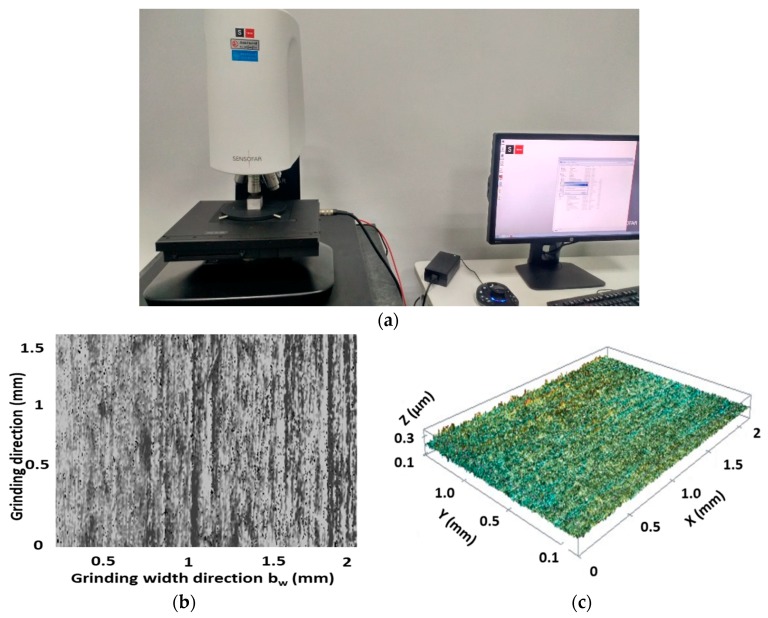
(**a**) General view of 3D optical profilometer S neox (Sensofar, Barcelona, Spain), (**b**) micrography of the ground surface, and (**c**) surface topography of the ground surface.

**Table 1 materials-11-02269-t001:** Chemical composition of AISI D2 Steel.

Element	C	Si	Mn	Cr	Mo	V	P	S	Ni	Fe
%	1.56	0.30	0.40	11.90	0.78	0.80	0.023	0.015	0.05	Balance

**Table 2 materials-11-02269-t002:** Process parameters and their levels for Dry, Wet, and MQL grinding.

SL. No.	Parameter (s)	−α	Low (−1)	Centre (0)	High (+1)	+α
Dry, Wet, MQL						
1	Depth of cut: $a_{p}$, (µm)	15	20	25	30	35
2	Table speed: $v_{w}$, (m/min)	3	5	7	9	11
3	Cutting Speed: $v_{s}$, (m/s)	15	20	25	30	35
For MQL only						
4	MQL flow rate: Q, (mL/h)	50	100	150	200	250

**Table 3 materials-11-02269-t003:** Design matrix for dry and wet condition grinding.

Exp. No.	Process Parameters			Response Measurements					
				Dry Grinding			Wet Grinding		
	$a_{p}$	$v_{w}$	$v_{s}$	*F*	*T*	$R_{a}$	*F*	*T*	$R_{a}$
	(µm)	(m/min)	(m/s)	(N)	(°C)	(µm)	(N)	(°C)	(µm)
1	20	5	20	82	245	0.5	60	69	0.29
2	30	5	20	115	270	0.58	70	83	0.34
3	20	9	20	83	240	1.4	54	80	0.58
4	30	9	20	117	291	1.51	81	84	0.63
5	20	5	30	85	245	1.12	46	58	0.48
6	30	5	30	92	297	0.89	71	94	0.46
7	20	9	30	110	248	1.45	47	71	0.59
8	30	9	30	103	310	1.49	80	92	0.51
9	15	7	25	82	225	0.81	40	62	0.51
10	35	7	25	119	319	1.07	90	109	0.56
11	25	3	25	86	268	0.46	55	74	0.25
12	25	11	25	105	300	1.6	65	82	0.65
13	25	7	15	111	262	1.01	61	70	0.40
14	25	7	35	98	276	1.4	49	80	0.55
15	25	7	25	90	262	1.35	56	69	0.42
16	25	7	25	91	260	1.39	55	70	0.43
17	25	7	25	90	268	1.42	54	68	0.41
18	25	7	25	93	260	1.41	58	71	0.42

**Table 4 materials-11-02269-t004:** Design matrix for MQL-assisted grinding.

Exp. No.	Process Parameters				Response Measurements		
	$a_{p}$	$v_{w}$	$v_{s}$	Q	F	T	$R_{a}$
	(µm)	(m/min)	(m/s)	(mL/h)	(N)	(°C)	(µm)
1	20	5	20	100	36	104	0.39
2	30	5	20	100	50	121	0.4
3	20	9	20	100	65	114	0.58
4	30	9	20	100	71	131	0.6
5	20	5	30	100	35	109	0.39
6	30	5	30	100	37	121	0.51
7	20	9	30	100	62	117	0.62
8	30	9	30	100	58	133	0.66
9	20	5	20	200	28	82	0.33
10	30	5	20	200	37	95	0.395
11	20	9	20	200	55	85	0.56
12	30	9	20	200	67	97	0.61
13	20	5	30	200	28	85	0.35
14	30	5	30	200	31	99	0.44
15	20	9	30	200	53	94	0.53
16	30	9	30	200	64	101	0.63
17	15	7	25	150	41	97	0.42
18	35	7	25	150	56	118	0.53
19	25	3	25	150	17	85	0.31
20	25	11	25	150	70	94	0.71
21	25	7	15	150	54	112	0.47
22	25	7	35	150	46	118	0.51
23	25	7	25	50	57	134	0.5
24	25	7	25	250	50	74	0.45
25	25	7	25	150	44	109	0.49
26	25	7	25	150	47	107	0.47
27	25	7	25	150	45	111	0.48
28	25	7	25	150	47	108	0.45
29	25	7	25	150	38	106	0.47
30	25	7	25	150	45	108	0.46

$v_{w}:$ Table speed; $a_{p}$: Depth of cut; $v_{s}$: Cutting speed; **Q**: MQL flow rate; $R_{a}$: Average surface roughness.

**Table 5 materials-11-02269-t005:** Grey relational coefficient (GRC) and grey relational grade (GRG) for MQL-grinding.

No.	GRC of Surface Roughness	GRC of Temperature	GRC of Normal Force	GRG
1	0.7143	0.5000	0.5870	0.6004
2	0.6897	0.3896	0.4500	0.5098
3	0.4255	0.4286	0.3600	0.4047
4	0.4082	0.3448	0.3333	0.3621
5	0.7143	0.4615	0.6000	0.5919
6	0.5000	0.3896	0.5745	0.4880
7	0.3922	0.4110	0.3750	0.3927
8	0.3636	0.3371	0.3971	0.3659
9	0.9091	0.7895	0.7105	0.8030
10	0.7018	0.5882	0.5745	0.6215
11	0.4444	0.7317	0.4154	0.5305
12	0.4000	0.5660	0.3506	0.4389
13	0.8333	0.7317	0.7105	0.7585
14	0.6061	0.5455	0.6585	0.6034
15	0.4762	0.6000	0.4286	0.5016
16	0.3846	0.5263	0.3649	0.4253
17	0.6452	0.5660	0.5294	0.5802
18	0.4762	0.4054	0.4091	0.4302
**19**	**1.0000**	**0.7317**	**1.0000**	**0.9106**
20	0.3333	0.6000	0.3375	0.4236
21	0.5556	0.4412	0.4219	0.4729
22	0.5000	0.4054	0.4821	0.4625
23	0.5128	0.3333	0.4030	0.4164
24	0.5882	1.0000	0.4500	0.6794
25	0.5263	0.4615	0.5000	0.4960
26	0.5556	0.4762	0.4737	0.5018
27	0.5405	0.4478	0.4909	0.4931
28	0.5882	0.4688	0.4737	0.5102
29	0.5556	0.4839	0.5625	0.5340
30	0.5714	0.4688	0.4909	0.5104

Bold indicates the optimum run.

**Table 6 materials-11-02269-t006:** ANOVA analysis for GRG of MQL grinding.

Source	SS	df	MS	*F*-Value	*p*-Value
Model	0.4686	14	0.0335	61.1807	<0.0001
Depth of Cut ($a_{p})$	0.0476	1	0.0476	86.9630	0.0001
Table Speed ($v_{w})$	0.2664	1	0.2664	487.0265	<0.0001
Cutting Speed ($v_{s})$	0.0011	1	0.0011	2.0556	0.1722
MQL Flow rate (Q)	0.0929	1	0.0929	169.8022	0.0001
$a_{p}\times v_{w}$	0.0054	1	0.0054	9.8734	0.0067
$a_{p}\times v_{s}$	0.0001	1	0.0001	0.2234	0.6432
$a_{p}\times Q$	0.0036	1	0.0036	6.6193	0.0212
$v_{w}\times v_{s}$	0.0001	1	0.0001	0.2026	0.6591
$v_{w}\times Q$	0.0032	1	0.0032	5.8058	0.0293
$v_{s}\times Q$	0.0003	1	0.0003	0.5101	0.4861
${a_{p}}^{2}$	0.0005	1	0.0005	0.9133	0.3544
${v_{w}}^{2}$	0.0359	1	0.0359	65.7029	0.0001
${v_{s}}^{2}$	0.0051	1	0.0051	9.3403	0.008
$Q^{2}$	0.0011	1	0.0011	2.0548	0.1722
Residual	0.0082	15	0.0005		
Lack of Fit	0.0071	10	0.0007	3.2625	0.102
Pure Error	0.0011	5	0.0002		
Cor. Total	0.4768	29			
Std. Dev.	0.023		R-Squared		0.9828
Mean	0.53		Adj R-Squared		0.9667
C.V. %	4.44		Pred R-Squared		0.9107
PRESS	0.043		Adeq Precision		30.121

**Table 7 materials-11-02269-t007:** Confirmation experiments.

Initial Cutting Conditions		Optimal Cutting Conditions	
		Predicted Results	Experimental Results
Levels	$a_{p}$^3^-$v_{w}$^1^-$v_{s}$^3^-Q^3^	$a_{p}$^1^-$v_{w}$^1^-$v_{s}$^3^-Q^5^	$a_{p}$^1^-$v_{w}$^1^-$v_{s}$^3^-Q^5^
Surface roughness (µm)	0.31		0.29
Temperature (°C)	85		81
Normal force (N)	17		16
GRG	0.9106	0.9242	0.9396
Improvement in GRG = 0.029; the % improvement in GRG = 3.18			

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
