# Peer review of "Multi-Objective Optimization for Grinding of AISI D2 Steel with Al2O3 Wheel under MQL"

_materials, 2018, doi:10.3390/ma11112269_

Reviewer 1 Report

The presented studies are very interesting and bring many important and significant results to the area of modern machining technologies. The article is carefully organized and well-illustrated. The presented issues don't raise substantive doubts. Noteworthy is the multithreading of carried out analyzes and the use of advanced mathematical mechanisms. I highly recommend this article to be accepted after the improve the following remarks:

1. Information given in Section 2.1 need improving and expanding. Please consider adding (e.g. in the tabular form) the characteristic of AISI D2 steel and technical designation of used grinding wheel (manufactured by Norton). Did the Authors verified the chemical composition of the workpiece material? If yes, I suggest to add a more obtained information.

2. Page 4, Figure 2: on schematic diagram a description of bottom part of an image is missing (stylus profilometer – left?, 3D optical profilometer right?). Please improve it.

3. Page 5, line 155: a full stop at the end of sentence is needed.

4. Page 5-6, Table 2-3: Ra or Ra ? Check the notation of  the parameter according to ISO 4287:1999 standard.

5. Page 8-9, Figure 4: there are no units characterizing the depth of cut and table speed on the charts. Please improve it.

6. Page 11, Figure 6: there are no units characterizing the depth of cut, table speed and MQL flow rate on the charts. Please improve it.

7. Page 11, line 287: should be “…portable stylus profilometer Surftest SJ-410 (Mitutoyo, Japan).”, rather than “Profilometer Surftest SJ-410”. The generally term profilometer is referred to both the contact (stylus) and non-contact (optical) instrument.

8. Page 13, line 329: see comment 4.

9. Page 13, Figure 8: there are no units characterizing the cutting speed and table speed on the charts. Please improve it.

10. Page 13-14, line 337-338 and 344: SENSOFAR – (Sensofar Metrology, Spain) is a producer of 3D optical profilometers. You used a one of these instruments – S NEOX (see Fig. 9a). Please correct text.

11. Page 13, line 340: should be “2 mm”, rather than “2mm”.

12. Page 14, line 345: should be “Figure 9”, rather than “Figure9”.

13. Page 14, Caption under Figure 9: My proposition: Figure 9. General view of 3D optical profilometer S NEOX (Sensofar, Spain) (a), micrography of the ground surface (b), surface topography of the ground surface (c).

14. Please expand the interpretation of results obtained from 3D optical profilometer.

Author Response

Response to reviewer 1

The presented studies are very interesting and bring many important and significant results to the area of modern machining technologies. The article is carefully organized and well-illustrated. The presented issues don't raise substantive doubts. Noteworthy is the multithreading of carried out analyzes and the use of advanced mathematical mechanisms. I highly recommend this article to be accepted after they improve the following remarks:

Response:

At beginning the authors acknowledge the efforts of the reviewer and at the same time thank him/her. The authors are highly pleased that the reviewer has considered the work interesting. At the same time, the reviewer has suggested some issues, which the authors believe if corrected would therefore increase the value of the article and make it more informative. The article has been modified according to the requirements of the reviewer, and the changed parts are marked with red color. Moreover, individual response has been pointed under each issue. Hope that it pleases the reviewer and elevates the scientific value of the article.

1. Information given in Section 2.1 need improving and expanding. Please consider adding (e.g. in the tabular form) the characteristic of AISI D2 steel and technical designation of used grinding wheel (manufactured by Norton). Did the Authors verified the chemical composition of the workpiece material? If yes, I suggest to add a more obtained information.

Reply: According to authors suggestions characteristic of AISI D2 steel and technical designation of used grinding wheel are added in the manuscript.

2. Page 4, Figure 2: on schematic diagram a description of bottom part of an image is missing (stylus profilometer – left?, 3D optical profilometer right?). Please improve it.

Reply: A description of bottom part of an image is added.

3. Page 5, line 155: a full stop at the end of sentence is needed.

Reply: Needful done.

4. Page 5-6, Table 2-3: Ra or Ra? Check the notation of the parameter according to ISO 4287:1999 standard.

Reply: Thanks for suggestion. The parameter is Ra, and it is corrected in the whole manuscript.

5. Page 8-9, Figure 4: there are no units characterizing the depth of cut and table speed on the charts. Please improve it.

Reply: Since the plots are generated by software, in the current condition, it is not possible to add the units of the parameters in the graph. However, with the title of the figure we have added the units of the parameters so that the readers can understand the measure.

6. Page 11, Figure 6: there are no units characterizing the depth of cut, table speed and MQL flow rate on the charts. Please improve it.

Reply: Since the plots are generated by software, in the current condition, it is not possible to add the units of the parameters in the graph. However, with the title of the figure we have added the units of the parameters so that the readers can understand the measure.

7. Page 11, line 287: should be “…portable stylus profilometer Surftest SJ-410 (Mitutoyo, Japan).”, rather than “Profilometer Surftest SJ-410”. The generally term profilometer is referred to both the contact (stylus) and non-contact (optical) instrument.

Reply: Needful included.

8. Page 13, line 329: see comment 4.

Reply: Needful corrected.

9. Page 13, Figure 8: there are no units characterizing the cutting speed and table speed on the charts. Please improve it.

Reply: Since the plots are generated by software, in the current condition, it is not possible to add the units of the parameters in the graph. However, with the title of the figure we have added the units of the parameters so that the readers can understand the measure.

10. Page 13-14, line 337-338 and 344: SENSOFAR – (Sensofar Metrology, Spain) is a producer of 3D optical profilometers. You used a one of these instruments – S NEOX (see Fig. 9a). Please correct text.

Reply: Needful modified.

11. Page 13, line 340: should be “2 mm”, rather than “2mm”.

Reply: Needful done.

12. Page 14, line 345: should be “Figure 9”, rather than “Figure9”.

Reply: Needful done.

13. Page 14, Caption under Figure 9: My proposition: Figure 9. General view of 3D optical profilometer S NEOX (Sensofar, Spain) (a), micrography of the ground surface (b), surface topography of the ground surface (c).

Reply: Caption under Figure 9, has been modified. Thanks for kind suggestion

14. Please expand the interpretation of results obtained from 3D optical profilometer.

Reply: Under Sec 3.4 the authors have already interpreted the results of 3D profilometer. Based on the data presented, it is difficult to expand any more details. To make such comments, it is required that more analysis is performed. Hence, our sincere apology is to you to disappoint you at this comment.

15. English language and style are fine/minor spell check required  

Reply: In preparing the revised manuscript, the authors have checked for possible mistakes in typos and spelling. However, due to different versions of MS Word, there, in some time, the spacing becomes a problem. We assure you that during the typesetting stage, that problem will be totally solved.

Reviewer 2 Report

The manuscript “Multi-objective optimization for grinding of AISI D2 2 steel with Al2O3 wheel under MQL” study the process parameters and response obtained for three different ways of grinding process.

The paper compares the dry, wet and MQL grinding. With this purpose a lot of experiments were carried out, modifing the depth of cut, table speed, cutting speed and flow rate for MQL process.

In my point of view, the paper is well structured. The introduction provides a wide compilation of the work made previously by others authors. The methods and experimental part is well explained and described, and the results are clear and very interesting. The conclusions resumes the work accomplished during the investigation.

I would like to recommend it for publication with the following minor considerations:

Line 31 – Abstract: Review the italics in the units.

Line 57 – Introduction: “Howeset al. [4] worked on the environmental effect of grinding fluids and reported that the grinding fluids have…”. Correct mistakes and revise the text.

Line 82 – Introduction: “by. [50].for” Correct mistakes.

Figure 1: Some parts of the figure (for example the legend) are not legible.

Line 136 – Experimental setup: “8.9 lbs/gal”. Review units.

Table 1: Review units format (italic).

Table 1 and 2: The nomenclature of the parameters is different.

Line 181 – Normal force: “only 22.33~55.27%”. The value does not match the one in figure 3

Line 277-278 – Temperature: “Figure 6bdepicts that in wet grinding the temperature 277 increases rapidly at a higher level of table speed. Figure4c represents the 3D…” Correct mistakes

Figure 7: In the legend of the third group of test there is a mistake.

Line 310 – Surface roughness: The authors claim that in some cases the MQL procedure is better than the wet one. It is a somewhat compromised statement, since only in one of the seven cases analyzed does this happen. This statement is repeated in the conclusions.

Line 345 – Surface topography…: “Figure9 indicates…” Correct mistakes.

Line 366 – Evaluation…: “usingeq. 13.” Correct mistakes.

Line 372 - Evaluation…: “factor?Correct mistakes.

References: Some references listed in this section are not referenced in the text of the paper.

Author Response

Response to reviewer 2

The manuscript “Multi-objective optimization for grinding of AISI D2 2 steel with Al2O3 wheel under MQL” study the process parameters and response obtained for three different ways of grinding process.

The paper compares the dry, wet and MQL grinding. With this purpose a lot of experiments were carried out, modifing the depth of cut, table speed, cutting speed and flow rate for MQL process.

In my point of view, the paper is well structured. The introduction provides a wide compilation of the work made previously by others authors. The methods and experimental part is well explained and described, and the results are clear and very interesting. The conclusions resume the work accomplished during the investigation.

I would like to recommend it for publication with the following minor considerations:

Response: At beginning the authors acknowledge the efforts of the reviewer and at the same time thank him/her. The authors are highly pleased that the reviewer has considered the work interesting. At the same time, the reviewer has suggested some issues, which the authors believe if corrected would therefore increase the value of the article and make it more informative. The article has been modified according to the requirements of the reviewer, and the changed parts are marked with red color. Moreover, individual response has been pointed under each issue. Hope that it pleases the reviewer and elevates the scientific value of the article.

Comment 1:

Line 31 – Abstract: Review the italics in the units

Reply: Needful is done.

Comment 2:

Line 57 – Introduction: “Howeset al. [4] worked on the environmental effect of grinding fluids and reported that the grinding fluids have…”. Correct mistakes and revise the text

Reply: Needful is not done. Due to different versions of MS Word, there, in some time, the spacing becomes a problem. We assure you that during the typesetting stage, that problem will be totally solved.

Comment 3:

Line 82 – Introduction: “by. [50].for” Correct mistakes

Reply: Needful modified.

Comment 4:

Figure 1: Some parts of the figure (for example the legend) are not legible

Reply: Figure 1 is modified.

Comment 5:

Line 136 – Experimental setup: “8.9 lbs/gal”. Review units.

Reply: The SI unit is used. The converted value is “1066 kg/m3” instead of “8.9 lbs/gal”.

Comment 6:

Table 1: Review units format (italic).

Reply: Needful done.

Comment 7:

Table 1 and 2: The nomenclature of the parameters is different.

Reply: Needful done.

Comment 8:

Line 181 – Normal force: “only 22.33~55.27%”. The value does not match the one in figure 3.

Reply: Thank you very much for highlighting mistake, needful is corrected i.e., “only 22.33~57.27%”.

Comment 9:

Line 277-278 – Temperature: “Figure 6b depicts that in wet grinding the temperature 277 increases rapidly at a higher level of table speed. Figure 4c represents the 3D…” Correct mistakes.

Reply: Needful is done.

Comment 10:

Figure 7: In the legend of the third group of test there is a mistake

Reply: Needful is done.

Comment 11:

Line 310 – Surface roughness: The authors claim that in some cases the MQL procedure is better than the wet one. It is a somewhat compromised statement, since only in one of the seven cases analyzed does this happen. This statement is repeated in the conclusions.

Reply: That is right authors have observed the better performance of MQL over WET in some cases. In Fig. 7 of the manuscript, author have compared the seven experiments with similar kind of experimental conditions, however, if the 4th parameter of MQL flow rate in MQL environment is not considered and comparison is drawn between only three cutting parameters, it can be clearly seen from following figure that MQL shows better results than Wet in some cases.

Comment 12:

Line 345 – Surface topography…: “Figure9 indicates…” Correct mistakes

Reply: Needful done.

Comment 13:

Line 366 – Evaluation…: “usingeq. 13.” Correct mistakes

Reply: Needful done. Due to different versions of MS Word, there, in some time, the spacing becomes a problem. We assure you that during the typesetting stage, that problem will be totally solved.

Comment 14:

Line 372 - Evaluation…: “factork” Correct mistakes

Reply: Needful done. Due to different versions of MS Word, there, in some time, the spacing becomes a problem. We assure you that during the typesetting stage, that problem will be totally solved.

Comment 15

References: Some references listed in this section are not referenced in the text of the paper.

Reply: All the papers are referred inside the manuscript. Please note that some references are referred in Figure 1.
